

# Dynamic coefficient symmetric polynomial-based secure key management scheme for Internet of Things (IoT) networks

Zhongya Liu[1],* and Yunxiao Luo[2],*

[1] Chongqing University of Posts and Telecommunications, Chongqing, China
[2] Chongqing Technology and Business Institute, Chongqing, China
* These authors contributed equally to this work.

## ABSTRACT

**Background:** With the extensive application and continuous expansion of the Internet of Things (IoT), the access of a large number of resource-limited nodes makes the IoT application face a variety of security vulnerabilities and efficiency limitations, and the operating efficiency and security of IoT are greatly challenged. Key management is the core element of network security and one of the most challenging security problems faced by wireless sensor networks. A suitable key management scheme can effectively defend against network security threats. However, among the key management schemes that have been proposed so far, most of them do not take into account the efficiency in terms of connectivity rate and resource overhead, and some of them even have security risks.

**Methods:** In this article, based on the symmetric polynomial algorithm, a dynamic coefficient symmetric polynomial key management scheme is proposed to better solve the IoT security problem. In this scheme, the nodes' *IDs* are mapped into the elements of the shared matrix *M* by the identity mapping algorithm, and these elements are used to construct polynomials *P(x,y)* to generate pairwise keys. The communicating nodes have their own coefficients of *P(x,y)* and thus have higher connectivity.

**Results:** The overall performance evaluation shows that the scheme significantly improves the resilience against node capture and effectively reduces the communication and storage overheads compared to the previous schemes. Moreover, the scheme overcomes the $\lambda$-security of symmetric polynomial key management scheme, and is able to provide a large pool of polynomials for wireless sensor networks, facilitating large-scale application of nodes.

# INTRODUCTION

In the network architecture of the Internet of Things (IoT), some new distributed models (*Bouarourou, Boulaalam & Nfaoui, 2021*; *Labib et al., 2021*) and multi-hop clustering scheme (*Muthukkumar et al., 2022*) for heterogeneous wireless sensor networks (WSNs) to

Corresponding author
Yunxiao Luo,
luoyunxiao@cqtbi.edu.cn

efficiently deal with heterogeneous sensors in a dynamic IoT environment, it greatly increases the service life and communication efficiency of node networkst. In the security architecture of the IoT, (*Alansari et al., 2023*) presents a novel lightweight system for anomaly detection of grayhole, blackhole, and selective forwarding attacks, and a survey (*Osamy et al., 2022*) reviews and analyzes the research trends related to the utilized artificial intelligence (AI) methods for IoT and the potential enhancement of IoT. In terms of key management, due to the complexity of the structure and high mobility of the IoT structure, the key management strategy based on static implementation cannot be directly applied to the IoT environment. In order to deal with the drawbacks of security and efficiency of the Internet of Things, the design of key management scheme has become one of the key breakthroughs to provide a secure and efficient IoT environment for key negotiation and secure communication between users. At the same time, compared with the common computer network key management model, the key management model oriented to IoT must meet the integrity, reliability, privacy and anonymity, non-repudiation and other characteristics, security and efficiency become the basic standards to measure its availability. Therefore, the IoT needs to design a more appropriate security key management technical solution to solve the security problem.

In order to reduce the energy overhead of wireless sensor networks in the IoT environment and improve network security, several security solutions have been proposed in the literature. *Blundo et al. (1998)* proposed a scheme in which all nodes use a common polynomial to establish a pairwise key. This simplifies the process of establishing a shared key and has a high connectivity rate between sensor nodes. In *Liu, Ning & Li (2003)*, the authors presented a protocol using grid-based key predistribution and random subset assignment. This scheme further extends the idea of a polynomial to establish pairwise keys. In random subset assignment, nodes select a set of polynomials from a pool and use the same polynomials in selected polynomials to establish pairwise keys. *Das & Sengupta (2008)* proposed a deterministic group-based key pre-distribution scheme based on a hierarchical wireless sensor network using bivariate polynomials over a finite field. *Zhang, Li & Li (2018)* developed another key predistribution for pairwise key establishment, which fuses polynomial pool-based and random key predistribution.

*Reegan & Baburaj (2017)* have proposed a triple key distribution scheme based on polynomial and multivariate mapping, aiming at the problem that the security performance of previous schemes decreases with the increase of the number of damaged nodes. And an improved energy-saving key distribution and management scheme has been proposed by *Chakavarika, Gupta & Chaurasia (2017)*. This scheme is scalable in terms of storage cost and computation cost. The security of network is improved by introducing encrypted random numbers in the key updating stage. In order to solve the memory overhead problem of q-composite scheme, *Gandino, Ferrero & Rebaudengo (2017)* proposed a new protocol q-s-composite to improve the efficiency of memory management. *Albakri & Harn (2019)* proposed a group key pre-distribution scheme based on probabilistic polynomials. Which significantly reduces the probability of sensor being attacked by node capture, and improves the security of wireless sensor networks. *Li et al. (2020)* proposed a secure random key distribution scheme aiming at the shortcomings of

q-s-composite scheme in literature (*Gandino, Ferrero & Rebaudengo, 2017*), which could not resist node replication attacks, combining localization algorithm with voting mechanism. To support the detection and cancellation of malicious nodes, and further modify parameters to resist node replication attacks. In order to solve the security of key management and IoT security performance index optimization problems, *Harn, Hsu & Xia (2021)* proposed a novel key distribution scheme. The key distribution protocol only needs logical XOR operation, which is much faster than other schemes. *Wang et al. (2020)* proposed a WSN layer-cluster key management scheme based on quadratic polynomial and Lagrange interpolation polynomial is proposed. *Nafi, Bouzefrane & Omar (2020)* proposed a new lightweight matrix-based key management protocol for IoT network. *Sharma & Purushothama (2022)* proposed a new and efficient scheme (BP-MGKM) for secure multi-group key management based on bivariate polynomial. *Najafi & Babaie (2023)* proposed approach, a lightweight hierarchical key management approach, generating shorter and more secure keys due to the use of a hierarchical structure based on the position and remaining energy of nodes. *Msolli et al. (2023)* proposed the key management scheme with pool-hash for the establishment, which exposes a new key pool contains original keys and other hashed admit the same identities thus new session keys transmitted in sensors nodes are established during the discovery and path key phases. *Taurshia et al. (2022)* presented a novel Group Key Management scheme for Low-Resource Devices (GKM-LRD) to offer key management service to groups in IoT applications. *Nafi et al. (2022)* proposed a new key management protocol aiming to secure communications before and after key establishment, used hash and one–one functions to achieve security during the key establishment process. *Rezaeipour & Barati (2022)* presented a key management protocol that delivers services such as message confidentiality, integrity, and authenticity to wireless sensor networks by handling keys generation, distribution, and maintenance. *Kandi et al. (2022)* proposed a novel decentralized blockchain-based protocol for the IoT, balancing the loads between nodes according to their capabilities. *Wei et al. (2021)* proposed two space-efficient Bitcoin-compatible key management schemes for the lighting network, based on the hash function and trapdoor one-way function, respectively.

However, for all these solutions, there are shortcomings in the overall performance of connectivity, energy consumption, and security. Especially, a small number of compromised nodes may affect the majority of pairwise keys. This greatly restricts the maximum number (nodes' limitation number) of nodes the sensor network can hold if the polynomial is unconditionally secure. While increasing the safety threshold $\lambda$ of the polynomial $f(x,y)$, these schemes are able to make the nodes' limitation number $\lambda$ a little enlarged. But it will make nodes suffer from some more serious problem, such that polynomials can extremely enhance the computation and storage overhead at the same time.

Making node communication of these schemes more secure requires a new mechanism to address the $\lambda$-secure problem. In addition, we should also consider the following questions: as some nodes are captured, the communication security of other nodes will be directly affected, and all nodes could even be compromised. It should be noted that the

energy, storage and communication overhead of sensor nodes are limited. The proposed scheme should maintain high node connectivity and low energy consumption.

Our scheme has better resilience to node capture attacks, a high connectivity rate between nodes, and can considerably reduce the communication and storage overhead. The contributions of this article are summarized as follows:

- Offers more effective resilience to node capture attacks for the λ-secure problem and has better resistance than other key management schemes.

- The wireless sensor network has a high connectivity rate. Our scheme uses an identity mapping algorithm to map a series of coefficients of $P(x, y)$, and each pair of communication nodes is able to establish a pairwise key.

- Low computational overhead. Even though the phase of pairwise key establishment consumes more energy than the other key management schemes, the node's chip is sufficient to deal with polynomial and hash algorithms.

- Low communication overhead. All communication nodes exchange identity information with each other, and the sensor network directly implements the identity mapping algorithm with the identity information of sensor nodes to get pairwise keys. There are no extra communication streams, except for identity information, during the process of pairwise key establishment, which can greatly reduce communication overhead.

- Low storage overhead. With a value of λ = 7, our scheme in pairwise key establishment generates quite a small amount of code. The shared matrix $M$ can generate a series of coefficients of $P(x, y)$ according to the sensor node's $ID$, and the sensor network of a head-cluster node is able to hold $1.75 \times 10^5$ nodes, which meets the requirements of many scenarios.

The structure of the article is as follows. "Preliminaries" describes the preliminaries. "Overview of Proposed Scheme" proposes our scheme with dynamic coefficient symmetric polynomials including key pre-distribution and key agreement phase. In "Theoretical Analysis", we analyze some classical security. "Performance Analysis" presents a comparative study and simulation results. "Conclusions" presents the conclusion of this article.

## PRELIMINARIES

In this section, we introduce the network model and background, as well as explain the symbols used and their descriptions in this article.

### Notation
Table 1 shows the notation used in the article.

### Network model
Our scheme is suitable for a distributed network architecture, which consists of remote server nodes, gateway nodes and sensor nodes. All sensor nodes in the network have the

**Table 1 The notation used in this article.**

| Notation | Description |
|---|---|
| $ID_{Nm}$ | Identity of node $Nm$ |
| $Hash()$ | *hash* function |
| $K_{lm}$ | The pairwise key between nodes $Nl$ and $Nm$ |
| $l_k$ | The length of key. Note that each generated key takes the same storage as a coefficient of the $\lambda$-degree polynomial. |
| $l_{ID}$ | The length of a node or key identifier |
| $q$ | $q$ is a prime number that is large enough to accommodate a cryptographic key, where $l_k = \log_2 q$ |
| $|\omega|$ | Size of polynomial pool |
| $\lambda$ | The security threshold |
| $N$ | The number of sensor nodes in a network |
| $\tau$ | The number of polynomials preloaded in each node |
| $s$ | The number of keys preloaded in each node |
| $\oplus$ | XOR operation |

same resources with the functions of sensing, collecting and transmitting data. Moreover, these sensor nodes are able to communicate with each other. Figure 1 shows the network model we have assumed.

## OVERVIEW OF PROPOSED SCHEME

In this section, we describe the key management scheme of dynamic coefficient symmetric polynomial for IoT networks, which has two phases: pairwise predistribution phase and pairwise key agreement phase. In the following, we describe the details of each phase.

### Key pre-distribution phase

(1) Before nodes are deployed, each node stores the information in advance, including a unique node *ID* and a hash function.

(2) Node $Nl$ ($l = 1, 2, \cdots, N$) is preload the shared matrix $M$. $N$ indicates the number of nodes in the network. The size of $M$ is $(\lambda + 1) \times (\lambda + 1)$, and element $a_{i,j}$ ($i, j = 0, 1, 2, \cdots, \lambda$) is over a finite field $F(q)$, where $q$ is a prime number that is large enough to accommodate a cryptographic key.

$$M = \begin{bmatrix} a_{0,0} & a_{0,1} & \cdots & a_{0,\lambda} \\ a_{1,0} & a_{1,1} & \cdots & a_{1,\lambda} \\ \vdots & \vdots & \ddots & \vdots \\ a_{\lambda,0} & a_{\lambda,1} & \cdots & a_{\lambda,\lambda} \end{bmatrix} \qquad (1)$$

### Key agreement phase

The pairwise key is used for end-to-end unicast communication. Communicating nodes authenticate each other's identity, construct the matrix coordinates $\{(w_i, v_j)|i, j = 0, 1, 2, \cdots, \lambda\}$ and coefficient set $\{a_{(w_i, v_j)}|i, j = 0, 1, 2, \cdots, \lambda\}$ of $P(x, y)$, and establish a pairwise key, as shown in Fig. 2.

**Peer**J Computer Science

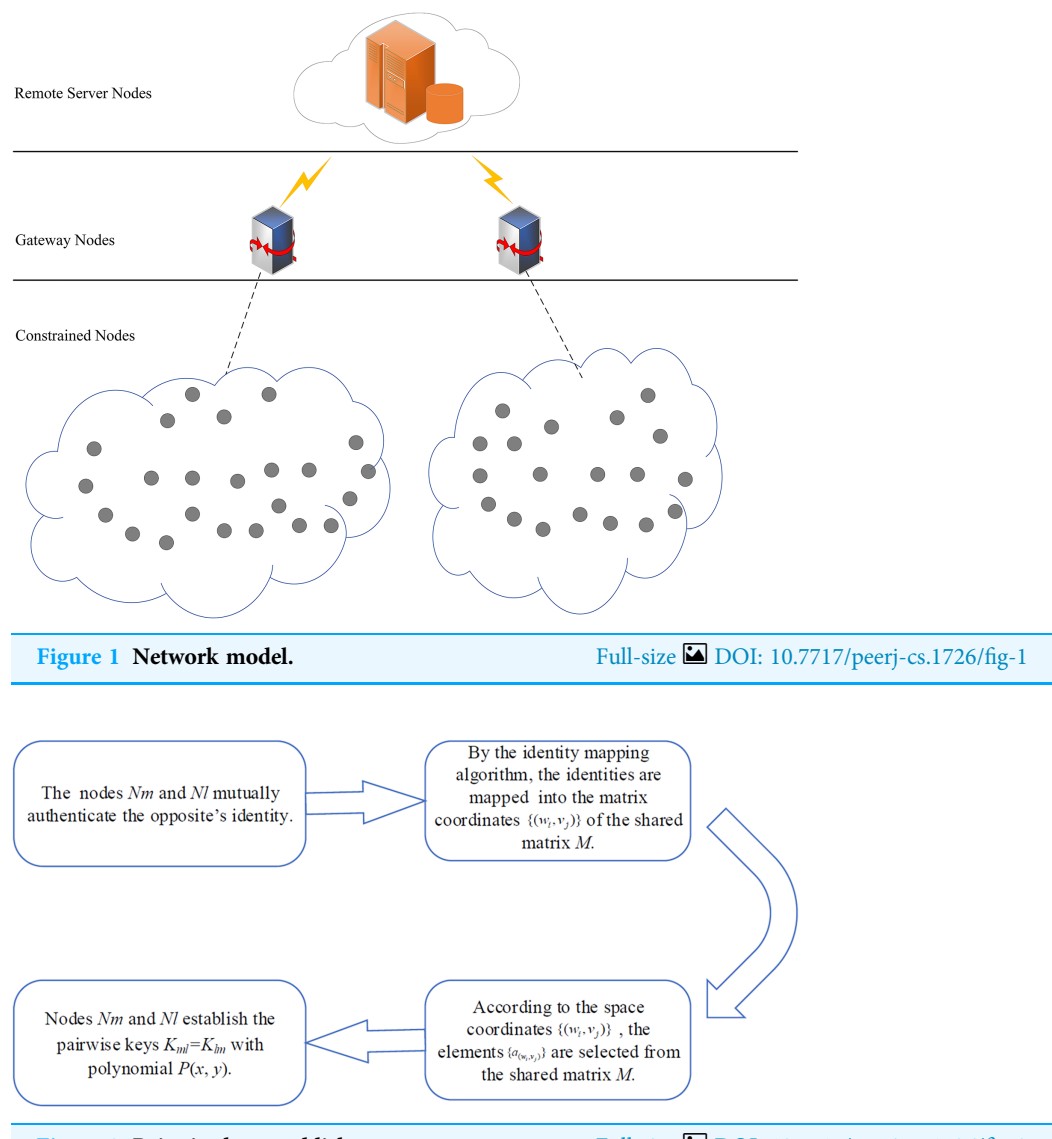

**Figure 1  Network model.**

**Figure 2  Pairwise key establishment.**

**Construct matrix coordinate set $C_{ml}$ and coefficient set $A_{ml}$.** Nodes $Nl$ and $Nm$ map their identities to matrix coordinates $(w_i, v_j)$ of $M$ and obtained the element $a_{(w_i, v_j)}$ from $M$ by the matrix coordinates $(w_i, v_j)$. Figure 3 shows the identity mapping algorithm.

(1) Nodes $Nl$ and $Nm$ obtain the mapped identity,

$$ID_{ml} = ID_{Nm} \oplus ID_{Nl}. \tag{2}$$

(2) Nodes $Nl$ and $Nm$ output more than $2t\,(\lambda + 1)$ bits of fixed-length data.

$$L = Hash(ID_{ml}) = w_0, w_1, \ldots, w_\lambda, v_0, v_1, \ldots, v_\lambda, \ldots \tag{3}$$

(3) The first $t(\lambda + 1)$ bits in $L$ are mapped to the row coordinate $w_i$ of $M$ as

$$W = w_0, w_1, \ldots, w_\lambda \tag{4}$$

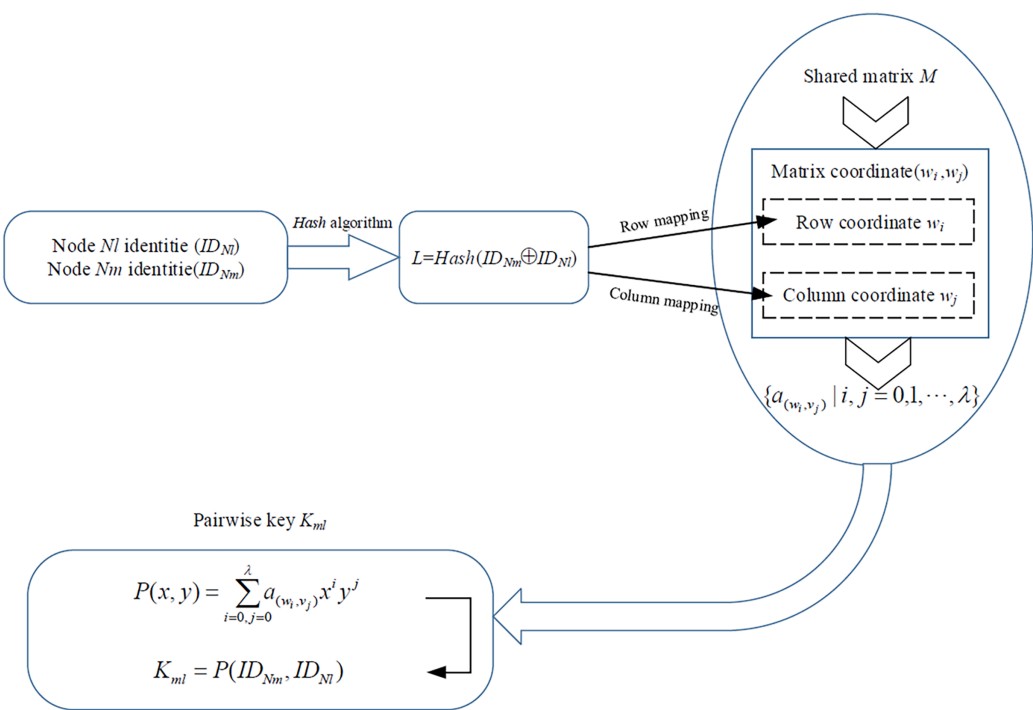

**Figure 3 Identity mapping algorithm.**

The length of $w_i$ $(0 \leq i \leq \lambda)$ is fixed at $t(t = 1, 2, \ldots)$ bits with $\lambda = 2^0 + 2^1 + \ldots + 2^{(t-1)}$, indicating the row coordinates of $M$. The output $L$ of the *hash* function is binary and must be converted to decimal form. For instance, $w_1 = 01011$ can be expressed in decimal form as $w_1 = 0 \times 2^4 + 1 \times 2^3 + 0 \times 2^2 + 1 \times 2^1 + 1 \times 2^0 = 11$.

(4) The next $t(\lambda + 1)$ bits in $L$ are mapped to column coordinate $v_j$ of $M$ as

$$V = v_0, v_1, \ldots, v_\lambda. \tag{5}$$

The length of $v_j$ $(0 \leq j \leq \lambda)$ is fixed at $t(t = 1, 2, \ldots)$ bits with $\lambda = 2^0 + 2^1 + \ldots + 2^{(t-1)}$, indicating the column coordinates of $M$. Like $w_i$, $v_j$ is binary and must be converted to decimal form.

(5) Nodes $Nl$ and $Nm$ generate the matrix coordinate set $C_{ml}$ of symmetric polynomial $P(x,y)$, with row coordinates $w_i$ and column coordinates $v_j$,

$$C_{ml} = \{(w_i, v_j) | i, j = 0, 1, \ldots, \lambda\}_{ml}. \tag{6}$$

The coefficient set $A_{ml}$ of $P(x,y)$ is established by $C_{ml,}$ and then used as coefficients of $P(x,y)$,

$$A_{ml} = \{a_{(w_i, v_j)} | i, j = 0, 1, \ldots, \lambda\}_{ml}. \tag{7}$$

**The communication nodes establish pairwise keys with $P(x,y)$.** Nodes $Nm$ and $Nl$ respectively construct the pairwise keys $K_{ml}$ and $K_{lm}$, as shown in Fig. 4, according to the elements $a_{(w_i, v_j)}$ of $M$, as

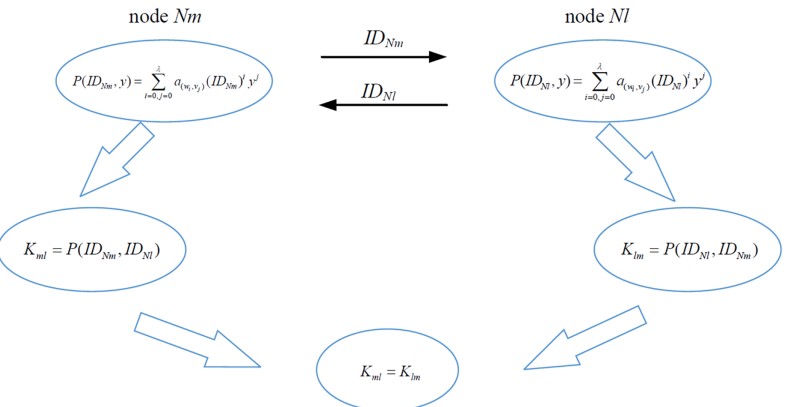

**Figure 4 Pairwise key establishment of nodes Nl and Nm.**

$$P(x, y) = \sum_{i=0,\, j=0}^{\lambda} a_{(w_i, v_j)} x^i y^j. \tag{8}$$

(1) When node $Nm$ executes $P(x,y)$, the input terms of the variable $x$ and $y$ are $x = ID_{Nm}$, $y = ID_{Nl}$,

$$\begin{aligned} K_{ml} &= P(ID_{Nm}, ID_{Nl}) \\ &= \sum_{i=0, j=0}^{\lambda} a_{(w_i, v_j)} (ID_{Nm})^i (ID_{Nl})^j. \end{aligned} \tag{9}$$

(2) When node $Nl$ executes $P(x, y)$, we input $x = ID_{Nl}$, $y = ID_{Nm}$, and

$$\begin{aligned} K_{lm} &= P(ID_{Nl}, ID_{Nm}) \\ &= \sum_{i=0, j=0}^{\lambda} a_{(w_i, v_j)} (ID_{Nl})^i (ID_{Nm})^j. \end{aligned} \tag{10}$$

(3) Since the polynomial $P(x,y)$ is symmetric, $K_{ml}$ is equal to $K_{lm}$, and

$$P(ID_{Nm}, ID_{Nl}) = P(ID_{Nl}, ID_{Nm}). \tag{11}$$

So that nodes $Nl$ and $Nm$ have established a pairwise key, *i.e.*, when node $Nm$ communicates with node $Nl$, they have a common coefficient set $A_{ml}$ for $P(x,y)$. The coefficient sets $A$ of $P(x,y)$ are different in distinguishing pairs of communicating nodes. For example, when communicating, two pairs of nodes ($Nm$, $Nl$) and ($Nm$, $Nd$) in sensor network have different coefficient sets $A_{ml}$ and $A_{md}$, *i.e.*, the attacker cannot use Lagrange interpolation to reconstruct $P(x,y)$.

## THEORETICAL ANALYSIS

In this section, we evaluate the $\lambda$-secure of symmetric polynomial and size of polynomial pool. Theoretical analysis shows that the proposed scheme can effectively address the $\lambda$-secure, and also has a large enough size of polynomial pool to resist brute force attacks.

### $\lambda$-security of symmetric polynomial

The key management scheme based on a symmetric polynomial uses the polynomial.

$$P(x,y) = \sum_{i=0,\, j=0}^{\lambda} a_{(w_i,v_j)} x^i y^j. \tag{12}$$

For Formula (12), when we obtain more than $(\lambda+1) \times (\lambda+1)$ IDs and its pairwise keys, this information can form a matrix equation.

$$\begin{bmatrix} w_{0,0} & w_{0,1} & \cdots & w_{0,\lambda} \\ w_{1,0} & w_{1,1} & \cdots & w_{1,\lambda} \\ \vdots & \vdots & \ddots & \vdots \\ w_{\lambda,1} & w_{\lambda,1} & \cdots & w_{\lambda,\lambda} \end{bmatrix} \begin{bmatrix} a_{0,0} \\ a_{0,1} \\ \vdots \\ a_{\lambda,\lambda} \end{bmatrix} = \begin{bmatrix} P_{0,0} \\ P_{0,1} \\ \vdots \\ P_{\lambda,\lambda} \end{bmatrix} \tag{13}$$

The coefficient matrix $W$ and augmented matrix $\overline{W} = (W, P)$ are seen as Eqs. (14) and (15). When $R(W) = R(\overline{W}) = \lambda + 1$, Eq. (19) has a unique solution, known from related theorems of linear equations.

$$W = \begin{bmatrix} w_{0,0} & w_{0,1} & \cdots & w_{0,\lambda} \\ w_{1,0} & w_{1,1} & \cdots & w_{1,\lambda} \\ \vdots & \vdots & \ddots & \vdots \\ w_{\lambda,1} & w_{\lambda,1} & \cdots & w_{\lambda,\lambda} \end{bmatrix} \tag{14}$$

$$\overline{W} = \begin{bmatrix} w_{0,0} & w_{0,1} & \cdots & w_{0,\lambda} & P_{0,0} \\ w_{1,0} & w_{1,1} & \cdots & w_{1,\lambda} & P_{0,1} \\ \vdots & \vdots & \ddots & \vdots & \vdots \\ w_{\lambda,1} & w_{\lambda,1} & \cdots & w_{\lambda,\lambda} & P_{\lambda,\lambda} \end{bmatrix} \tag{15}$$

For a hierarchical network with $d$ nodes sharing one polynomial $P(x,y)$, every pair of communication nodes shares the same $(\lambda+1)^2$ coefficients selected from $M$, thus establishing $d$ pairwise keys. In theory, reconstructing $P(x,y)$ in $\lambda$-degree by Lagrange interpolation requires at least $(\lambda+1)$ nodes' information containing $(\lambda+1)$ polynomial values and $(\lambda+1)$ *ID*s. Lagrange interpolation is formulated as

$$P(x) = \sum_{j=0}^{\lambda} y_j \prod_{k=0,\, k \neq j}^{\lambda} \frac{x - x_k}{x_j - x_k}. \tag{16}$$

Obviously, with fewer than $\lambda$ sensor nodes, the attacker obtains insufficient information to reconstruct $P(x,y)$ by Lagrange interpolation. The proposed key management scheme maps the *ID*s of communication nodes to different elements $a_{ij}$ ($i, j = 0, 1, \ldots, \lambda$) in $M$ by

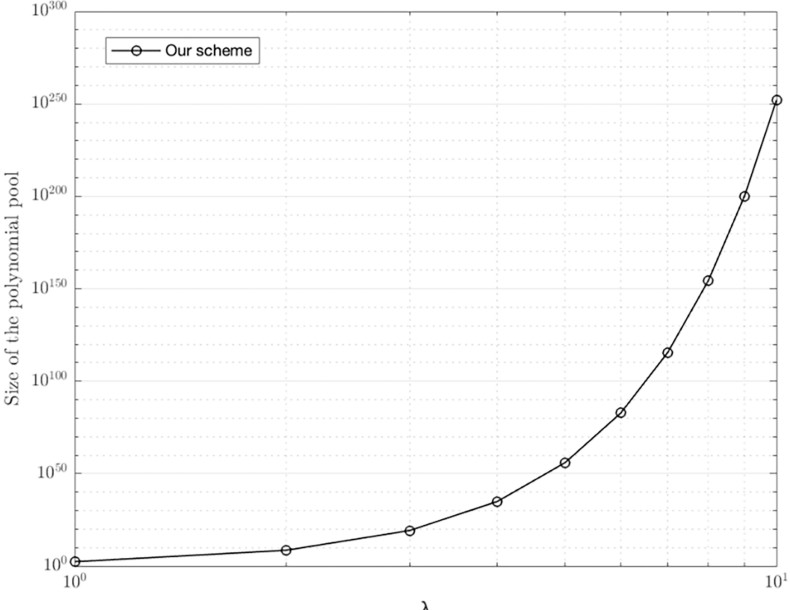

**Figure 5 Size of polynomial pool with different λ.** 

identity mapping algorithm, and uses these elements $a_{ij}$ ($i, j = 0, 1, \ldots, \lambda$) as coefficients of $P(x,y)$. Since every node has a unique *ID*, the selected elements $a_{ij}$ ($i, j = 0,1, \ldots, \lambda$) are also different. Hence it is difficult for attackers to reconstruct $P(x,y)$ by Lagrange interpolation.

## Size of polynomial pool with different $\lambda$

Due to the limitation of $\lambda$ and $t$, our scheme uses the hash function of length is $2t(\lambda +1)$ bits, which makes coefficients to be combined $[(\lambda + 1)^2]^{(\lambda+1)^2}$ polynomials. That is, our key management scheme generates a polynomial pool $\omega$ containing large

$|\omega| = [(\lambda + 1)^2]^{(\lambda+1)^2}$ polynomials, and each node selects a polynomial from the polynomial pool. In other words, every node in the network has the probability $1/|\omega|$ to carry a polynomial from the polynomial pool. Given the security, our scheme should put a limit on the maximum $|\omega|(\lambda + 1)^2$ nodes. Figure 5 shows the size of polynomial pool $|\omega|$ with different λ. When λ = 3, our scheme can generate a polynomial pool holding 1.84 × $10^{19}$ polynomials, allowing our scheme to be used in most scenarios. In addition, our scheme is able to well resist brute force attacks. If someone wants to obtain a pairwise key, they should obtain the coefficients $a_{(w_i, v_j)}$ of $P(x,y)$ from the shared matrix $M$. There are a total of $[(\lambda + 1)^2]^{(\lambda+1)^2}$ possibilities for an attacker to repeat. When λ = 3, our scheme can generate a polynomial pool holding 1.84 × $10^{19}$ polynomials, where it's hard for an attacker to repeat so many times.

## PERFORMANCE ANALYSIS

We have conducted a comparative study of a number of related schemes, carried out scientific research, collected data through compliant methods, and evaluated the performance of our scheme through realistic simulations, including the resilience of node

capture, connectivity, and resource overhead (energy, memory, and computation) in the pairwise key establishment process.

## Resilience against node capture

A proper key management scheme should resist attacks while the network continues its normal operation. The main security threat to the proposed solution is the $\lambda$-secure. The security analysis of the new scheme proposed in this section mainly analyzes its resilience to node capture. Resilience is computed as the fraction of links compromised in non-compromised nodes. When the number of nodes is large enough in the network, the output values of *hash* functions will collide, and communication nodes will hold the same output value $L$ of the *hash* function and $P(x,y)$. Security analysis shows that our scheme not only offers more effective resilience to node capture attacks for the $\lambda$-secure, but has better resistance to node capture attacks compared with the other key management scheme.

### *Probability of at least one matrix being broken*

Denote that $S_i$ is an event in the *i*th polynomial $P(x,y)$ is cracked ($i \in \{1, 2, \ldots, |\omega|\}$). All coefficients $a_{ij}$ ($i, j = 0, 1, \ldots, \lambda$) selected from $M$ can be combined into $|\omega|$ different kinds of polynomials. The probability that one of these polynomial $P(x,y)$ ($P \in \{P_1, P_2, \ldots, P_{|\omega|}\}$) occurrence in a node is $\theta = \dfrac{1}{|\omega|}$. $Cx$ is an event that $x$ nodes are compromised in a network.

$$|\omega| = [(\lambda + 1)^2]^{(\lambda+1)^2}. \tag{17}$$

We have,

$$P_r(\text{atleast one } P(x, y) \text{ is broken}|C_x) = P_r(S_1 \cup S_2 \cup \ldots \cup S_{|\omega|}|C_x). \tag{18}$$

It is obtained by union bound,

$$P_r(S_1 \cup S_2 \cup \ldots \cup S_{|\omega|}|C_x) \leq \sum_{i=1}^{|\omega|} P_r(S_i|C_x). \tag{19}$$

It is equal probability for each $P(x, y)$ to be broken. That is,

$$\sum_{i=1}^{|\omega|} P_r(S_i|C_x) = |\omega| \cdot P_r(S_i|C_x). \tag{20}$$

Therefore,

$$P_r(\text{atleast one } P(x, y) \text{ is broken}|C_x) \leq \sum_{i=1}^{|\omega|} P_r(S_i|C_x) = |\omega| \cdot P_r(S_i|C_x) \tag{21}$$

The total number of terms in $P(x,y)$ is $n = (\lambda + 1)^2 = (2^0 + 2^1 + \ldots + 2^{(t-1)} + 1)^2$. Because every coefficient $a_{ij}$ ($i, j = 0, 1, \ldots, \lambda$) in $P(x,y)$ is different, the safety threshold of the proposed scheme is $\lambda' = n - 1 = (\lambda + 1)^2 - 1$. We have,

$$P_r(S_i|C_x) = \sum_{i=\lambda'+1}^{x} \binom{x}{i} \theta^i (1-\theta)^{x-i}. \tag{22}$$

Therefore we get the following upper bound:

$$P_r(\text{atleast one } P(x,y) \text{ is broken}|C_x) \leq \sum_{i=\lambda'+1}^{x} \binom{x}{i} \theta^i (1-\theta)^{x-i}$$
$$= \sum_{i=\lambda'+1}^{x} \binom{x}{i} \left(\frac{1}{|\omega|}\right)^i \left(1 - \frac{1}{|\omega|}\right)^{x-i}. \tag{23}$$

### The fraction of compromised network communication

$c$ is a link that two uncompromised nodes establish a communication with a common key $K$. $Bi$ is a event that one common key $K$ of two nodes is derived by compromised key space $S_i$.

$$P_r(c \text{ is broken}|C_x) = P_r(B_1 \cup B_2 \cup \ldots \cup B_{|\omega|}|C_x). \tag{24}$$

Since the link $c$ is built securely by one common key $K$ that derived by a key space $S_i$. Due to events $B_1, B_2, \ldots, B_{|\omega|}$ are mutually exclusive and probability of occurrence in $Bi$ is equally, therefore,

$$P_r(c \text{ is broken}|C_x) = \sum_{i=1}^{|\omega|} P_r(B_i|C_x) = |\omega| \cdot P_r(B_1|C_x). \tag{25}$$

Note that $(K \in S_1)$ represents an event that "key $K$ was derived by $S_i$."

$$P_r(B_1|C_x) = \frac{P_r((K \in S_1) \cap (S_1 \text{ is comprised}) \cap C_x)}{P_r(C_x)}. \tag{26}$$

Event $(K \in S_1)$ is independent of the events $Cx$ and ($S_1$ is compromised).

$$P_r(B_1|C_x) = \frac{P_r(K \in S_1) \cdot P_r(S_1 \text{ is comprised}) \cap C_x)}{P(C_x)}.$$
$$= P_r(K \in S_1) \cdot P_r(S_1 \text{ is comprised}|C_x) \tag{27}$$

The probability of event $(K \in S_1)$ is equal to the probability of event "the link $c$ established by space $S_i$". Each key space appears randomly and uniformly in a node.

$$P_r(K \in S_1) = P_r(\text{the link } c \text{ established by space } S_i) = \frac{1}{|\omega|}. \tag{28}$$

Therefore, the probability of event that one link is compromised when $x$ nodes captured show as

$$P_r(c \text{ is broken}|C_x) = |\omega| \cdot P_r(B_1|C_x)$$

$$= |\omega| \cdot \frac{1}{|\omega|} \cdot P_r(S_1 \text{ is comprised}|C_x)$$

$$= \sum_{i=\lambda'+1}^{x} \binom{x}{i} \left(\frac{1}{|\omega|}\right)^i \left(1 - \frac{1}{|\omega|}\right)^{x-i} \tag{29}$$

Assume that $w$ secure communication links do not involve any of the $x$ compromised nodes. Denote $R$ as the event "the rest of $w$ secure communication links except $x$ compromised nodes participating in". That is,

$$P_r(R) = \frac{w \cdot P_r(c \text{ is broken}|C_x)}{w}$$

$$= P_r(c \text{ is broken}|C_x) \tag{30}$$

The above equation indicates that, given that $x$ nodes are compromised, the fraction of the compromised secure communication links outside of those $x$ compromised nodes is the same as the probability of one $P(x, y)$ being compromised.

### Comparison to previous work

Although many key managements based on Blundo's scheme can ensure that all nodes are able to be connected, they are faced with $\lambda$-secure. Security analysis shows that our scheme not only offers more effective resilience to node capture attacks for the $\lambda$-secure, but has better resistance to node capture attacks compared with other key management schemes. Table 2 lists some different schemes of comparison used in this article.

Our scheme offers better resilience to node capture attacks than q-composite, *Liu, Ning & Li (2003)* and *Zhang, Li & Li (2018)* in Fig. 6. If attackers capture 231 sensor nodes, about 22.8% of the pairwise keys, in our scheme with $\lambda = 1$ and $t = 1$, between non-compromised nodes will be compromised. If an attacker captures 401 nodes, about 46.5% of the pairwise keys between non-compromised nodes will be compromised. In *Blundo et al. (1998)*, when captured nodes remain at 410, all pairwise keys between non-compromised nodes will be compromised. In *Liu, Ning & Li (2003)*, when captured nodes remain at 500, all pairwise keys between non-compromised nodes will be compromised. In *Zhang, Li & Li (2018)*, when captured nodes remain at 500, about 91.0% of pairwise keys between non-compromised nodes will be compromised. In q-composite ($q = 2$) and q-composite ($q = 3$), when captured nodes remain at 800, almost all pairwise keys between non-compromised nodes will be compromised.

### Resilience to node capture in our scheme

The relationship between $\lambda$ and $t$ can be described as $\lambda = 2^0 + 2^1 + \ldots + 2^{(t-1)}$ in our scheme, *i.e.*, $\lambda$ and $t$ affect the fraction of compromised links. Figure 7 shows the relationship between the fraction of compromised links for non-compromised sensors and the number of compromised nodes with different values of $\lambda$ and $t$. If attackers capture 1,000 sensor nodes, about 90.2% of the pairwise keys between non-compromised nodes

**Table 2 Some different schemes of comparison.**

| Scheme | Liu, Ning & Li (2003) | q-composite scheme (Chan, Perrig & Song, 2003) | | Our scheme | Zhang, Li & Li (2018) |
|---|---|---|---|---|---|
| s | – | 20 | | – | 12 |
| |S| | – | 340 | 200 | – | – |
| q | – | 2 | 3 | – | – |
| τ (2 ≤ τ < |ω|) | 2 | – | | – | 2 |
| |ω| | 11 | – | | – | 18 |
| t | – | – | | 1 | – |
| λ | 18 | – | | 1 | 18 |
| N | – | – | | – | 200 |

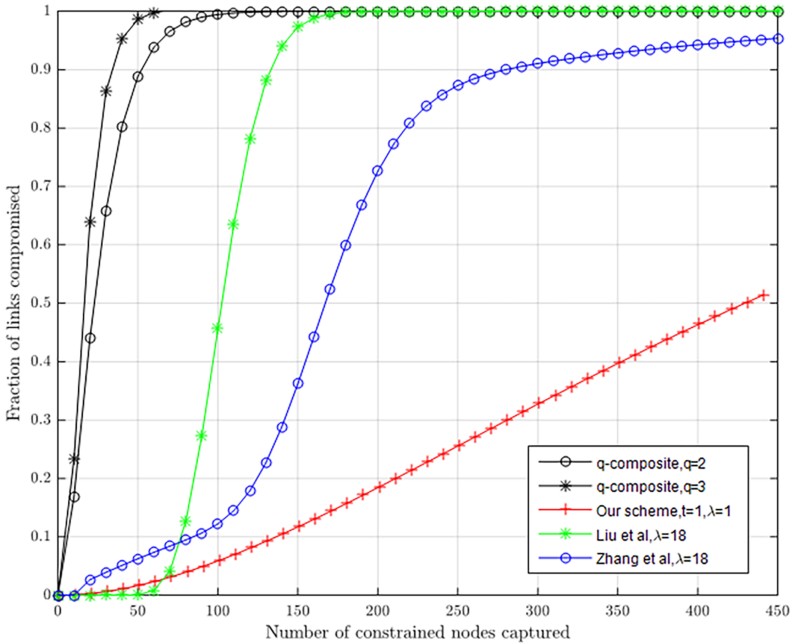

**Figure 6 Resilience to node capture attack.**

will be compromised when $\lambda = 1$ and $t = 1$. If attackers capture $1.0 \times 10^8$ sensor nodes, nearly no pairwise keys between non-compromised nodes will be compromised when $\lambda = 3$ and $t = 2$. Actually, with $\lambda = 3$ and $t = 2$, the sensor network can hold at least $1.0 \times 10^8$ nodes, which is enough to meet the requirements of many scenarios.

## Connectivity rate

A higher connectivity rate of the node network can increase the communication efficiency of the node and reduce the loss of communication energy. Figure 8 compares the network connectivity of the proposed scheme with that of *Liu, Ning & Li (2003)* and *q*-composite (*Chan, Perrig & Song, 2003*). In the simulation, we assume that the *q*-composite preloaded $s = 50$ keys in each sensor, and $\tau = 10$ and $\tau = 20$ polynomials in *Liu, Ning & Li (2003)* are

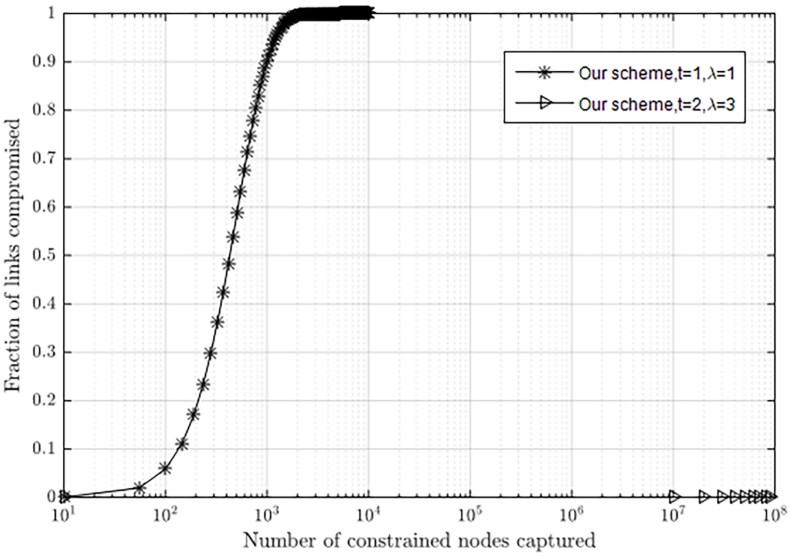

**Figure 7 Resilience to node capture attack with different λ and t.**

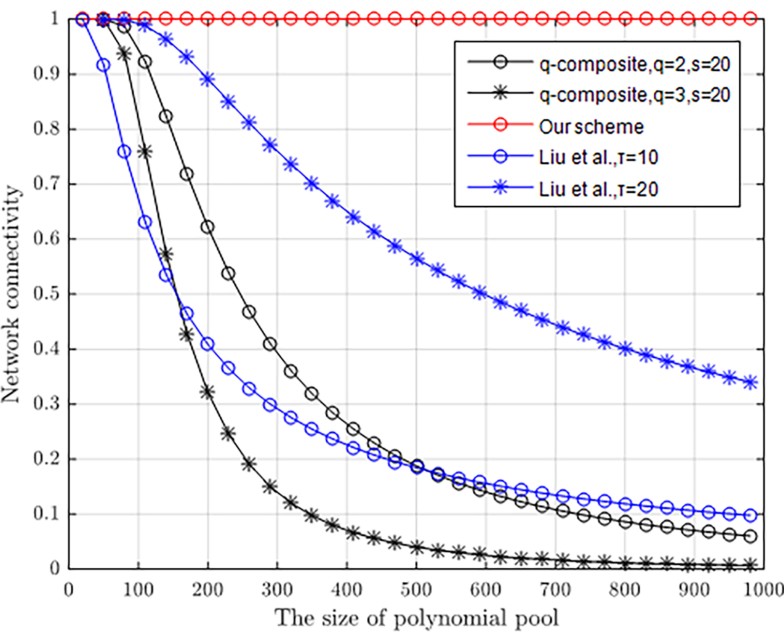

**Figure 8 Network connectivity with the different size of polynomial pool.**

preloaded in each node. The result illustrates that the proposed scheme has 100% connectivity regardless of the size of polynomial pool.

Table 3 in more detail shows that our scheme offers a better connectivity rate of nodes than the other three schemes, and inherits that every node in a sensor network is able to establish a connection in *Blundo et al. (1998)*. When three polynomials are selected from one polynomial pool holding 25 polynomials, about 33.0% of nodes in *Liu, Ning & Li*

**Table 3  Comparison of connectivity among various schemes.**

| Scheme | Connectivity $P$ |
|---|---|
| $q$-composite (*Chan, Perrig & Song, 2003*) ($q = 2$, $s = 20$, $|S| = 340$) | 0.332 |
| $q$-composite (*Chan, Perrig & Song, 2003*) ($q = 3$, $s = 20$, $|S| = 200$) | 0.320 |
| *Blundo et al. (1998)* ($\lambda = 18$) | 1 |
| *Blundo et al. (1998)* ($\lambda = 19$) | 1 |
| *Li et al. (2020)* ($\lambda = 18$, $\tau = 2$, $|\omega| = 11$) | 0.346 |
| *Li et al. (2020)* ($\lambda = 18$, $\tau = 3$, $|\omega| = 25$) | 0.330 |
| *Zhang, Li & Li (2018)* ($\lambda = 18$, $\tau = 2$, $s = 12$, $|\omega| = 18$, $N = 200$) | 0.318 |
| *Zhang, Li & Li (2018)* ($\lambda = 18$, $\tau = 3$, $s = 23$, $|\omega| = 18$, $N = 200$) | 0.578 |
| Our scheme ($t = 2$, $\lambda = 3$) | 1 |
| Our scheme ($t = 3$, $\lambda = 7$) | 1 |

*(2003)* are connected. However, when still remaining 34.6% of nodes are connected, *Liu, Ning & Li (2003)* should select two polynomials from a polynomial pool with 11 polynomials. In $q$-composite ($q = 2$) (*Chan, Perrig & Song, 2003*), when 20 keys are selected from one key pool holding 340 keys, about 33.2% of nodes are connected. However, when still remaining 32.0% of nodes are connected, the $q$-composite ($q = 3$) scheme (*Chan, Perrig & Song, 2003*) should scale down the key pool size to 200. When three polynomials are selected from one polynomial pool holding 18 polynomials, about 57.8% of nodes in *Zhang, Li & Li (2018)* are connected. However, when two polynomials are selected from the same polynomial pool, only about 31.8% of nodes in *Zhang, Li & Li (2018)* are connected.

## Resource overhead

We selected several classic key management solutions for comparative study from the aspects of communication, computation, and storage overhead. For convenience, we only consider the polynomial $P(x,y)$ cost. The $l_k$ represents the length of key. we assume that each generated key takes the same storage as the coefficient of a $\lambda$-degree polynomial. $l_{ID}$ is the length of a node or key identifier. We assume that the node and key identifiers have the same length. $\tau$ are the number of polynomials selected for each node in *Liu, Ning & Li (2003)* and *Zhang, Li & Li (2018)*. $s$ is the number of keys preloaded in each sensor.

We estimate the computation, communication, and storage energy consumed by constrained nodes during pairwise key establishment. Owing to selecting a set of polynomials from a polynomial pool, we use the same amount of polynomials for *Liu, Ning & Li (2003)* and *Zhang, Li & Li (2018)* per node. There are the same $\lambda$-degree for a polynomial for the four schemes. Table 4 lists the comparative study in resource overheads.

### Computational overhead

The key management schemes with polynomials rely on the existence of the same polynomial between two nodes. In other words, the polynomial has an important impact on the computation cost in nodes, when the pairwise key is established. To evaluate the

**Table 4 Comparative study in resource overheads.**

| Scheme | Storage overhead | Communication overhead | Computational overhead |
|---|---|---|---|
| *Blundo et al. (1998)* | $(\lambda+1)l_k + l_{ID}$ | $l_{ID}$ | $\lambda+1$ |
| *Liu, Ning & Li (2003)* | $[\tau(\lambda+1)]l_k + (\tau+1)l_{ID}$ | $(\tau+1)l_{ID}$ | $\lambda+1$ |
| *Zhang, Li & Li (2018)* | $[s+\tau(\lambda+1)]l_k + (2s+\tau+1)l_{ID}$ | $(2s+\tau+1)l_{ID}$ | $\lambda+1$ |
| Our scheme | $(\lambda+1)^2 l_k + l_{ID}$ | $l_{ID}$ | $\lambda+1$ |

**Table 5 Computational overhead of comparative study.**

| Scheme | *Blundo et al. (1998)* | *Liu, Ning & Li (2003)* | *Zhang, Li & Li (2018)* | Our scheme |
|---|---|---|---|---|
| Computational overhead | $\lambda+1$ | $\lambda+1$ | $\lambda+1$ | $\lambda+1$ |

**Table 6 Communication overhead of comparative study.**

| Scheme | *Blundo et al. (1998)* | *Liu, Ning & Li (2003)* | *Zhang, Li & Li (2018)* | Our scheme |
|---|---|---|---|---|
| Communication overhead | $l_{ID}$ | $(\tau+1)l_{ID}$ | $(2\tau+1)l_{ID}$ | $l_{ID}$ |

computational cost, we mainly consider the number of calculations in polynomial. Table 5 lists the computational overhead of the comparative study, which shows that these four schemes have the same consumption in computation. In fact, although these schemes owe the same computational overhead, our scheme has better advantages in applying to some field environments compared with the other three schemes, with better resilience against node capture and lower storage overhead.

### Communication overhead

The information exchange of wireless sensor networks relies on the emission of electromagnetic waves, which depletes much energy carried by nodes. In other words, energy cost in communication is significantly impacted by the length of communication message and the number of data packets. The longer the data length, the more energy in communication is consumed. These schemes mainly send the identities of the node, the identities of the secret key and the identities of the polynomial when the pairwise key is established. $\tau$ is the number of polynomials selected for each node. Table 6 lists the communication overhead of the comparative study, where $s = \frac{1}{2}\tau$.

Note that both $l_k$ and $l_{ID}$ are 32 bits. Figure 9 shows the comparison in communication consumption of our scheme with *Blundo et al. (1998)*, *Liu, Ning & Li (2003)* and *Zhang, Li & Li (2018)*, where the loss of energy in communication is represented by the length of information (bits). The figure clearly illustrates that the communication cost of our scheme is significantly lower than in *Zhang, Li & Li (2018)* and *Liu, Ning & Li (2003)*, and equal to *Blundo et al. (1998)* during pairwise key establishment. As we know, our scheme, similar to *Blundo et al. (1998)*, only needs nodes to pass their own identities to each other, which

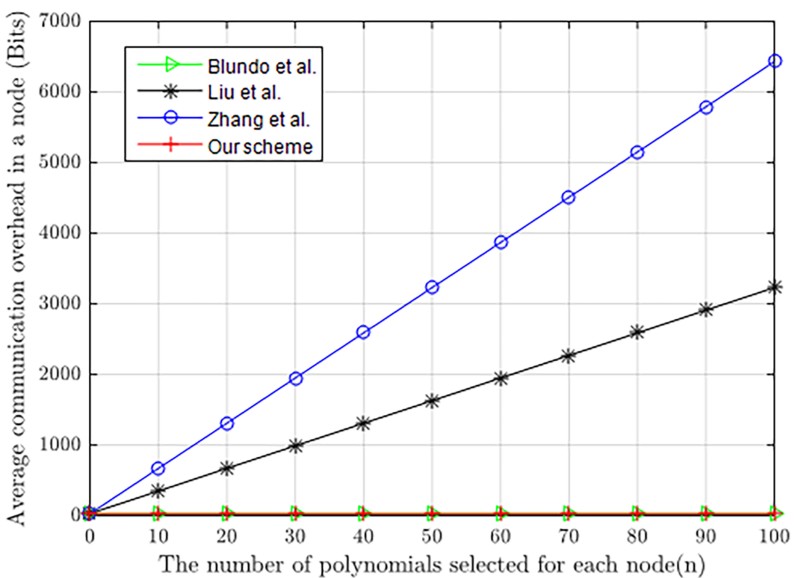

**Figure 9 Communication energy consumed by constrained nodes.**

**Table 7 Storage overhead of comparative study.**

| Scheme | Blundo et al. (1998) | Liu, Ning & Li (2003) | Zhang, Li & Li (2018) | Our scheme |
|---|---|---|---|---|
| Storage overhead | $(\lambda+1)l_k +l_{ID}$ | $50(\lambda+1)l_k+51l_{ID}$ | $[25+50(\lambda+1)]l_k+101l_{ID}$ | $(\lambda+1)^2 l_k +l_{ID}$ |

**Note:**
$s = 25$

improves communication efficiency and reduces the energy consumption of nodes in communication.

### Storage overhead

Sensor nodes are highly constrained in terms of memory resources. When designing a key management scheme, we should reduce the memory overhead of nodes as much as possible. The storage overhead this scheme depends on the cost of the nodes' dentities, the coefficients of the $\lambda$-degree polynomials and the dentities of polynomials, where $l_k = \log_2 q$. Table 7 lists the storage overhead of the comparative study, in which $\tau = 50$, $s = 25$.

Assume that both $l_k$ and $l_{ID}$ are 32 bits. Figure 10 shows the comparison in memory consumption (bits) of our scheme with *Blundo et al. (1998)*, *Liu, Ning & Li (2003)* and *Zhang, Li & Li (2018)*, which clearly shows the advantage of our scheme to some extent. As the $\lambda$ no more than 50, the storage cost in our scheme is much smaller than in *Zhang, Li & Li (2018)* and *Liu, Ning & Li (2003)* during pairwise key establishment. The storage overhead in our scheme remains stable, not changing with the number of nodes. In fact, with a small $\lambda$, our scheme can still hold a large number of nodes and maintain better resilience to node capture. Our scheme, when $\lambda = 3$, can hold $5.53 \times 10^{19}$ nodes in a network. However, *Liu, Ning & Li (2003)* ($\lambda = 18$, $\tau = 3$, $|\omega| = 25$) is only able to hold 158 nodes in a network. When $1.0 \times 10^8$ nodes are captured in a network, the capture rate of

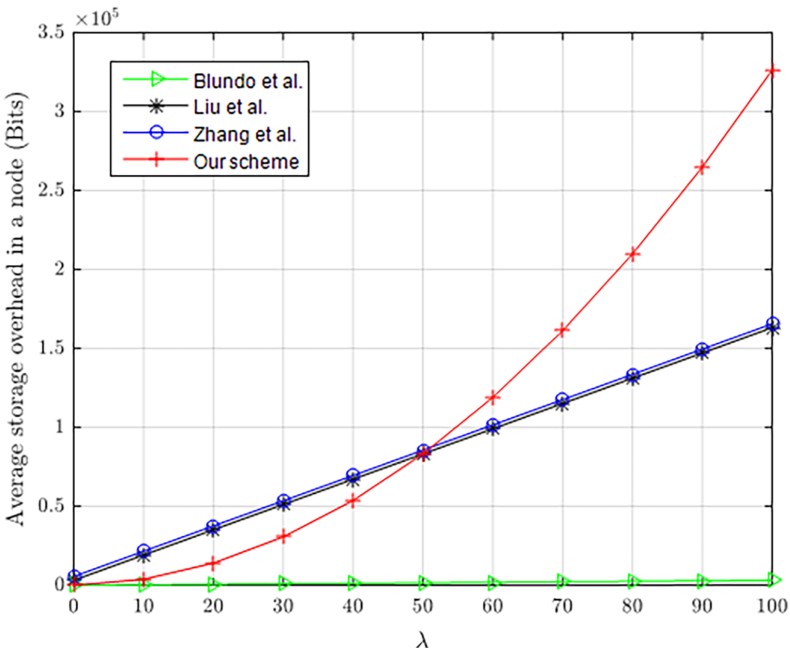

**Figure 10 Average memory consumption of constrained nodes.**

communication links in non-compromised nodes is still nearly close to 0. Therefore, our scheme has greater advantages in applying to large-scale networks, not only with lower storage overhead but also resilience to node capture attacks.

## CONCLUSIONS

We proposed a key management scheme with a dynamic coefficient symmetric polynomial. The scheme allows every pair of communicating nodes to use their own *ID*s to map into the elements of the shared matrix $M$ and assigns these elements to polynomial $P(x, y)$ to establish pairwise keys. Different from other schemes, ours deterministically configures a different polynomial for each pair of communicating nodes by an identity mapping algorithm. This enables a high connectivity rate and solves the λ-secure problem of key management, when no hash collision occurs. Security analysis shows that the proposed scheme has stronger resilience to node capture from various types of attacks. It consumes less energy in storage and communication than other protocols.

### Funding

This study was supported by the Science and Technology Research Program of Chongqing Municipal Education Commission C: (Grant: KJQN202104004) and the Natural Science Foundation of Chongqing (Grant: CSTB2022NSCQ-MSX1632). The funders had no role in study design, data collection and analysis, decision to publish, or preparation of the manuscript.

## Grant Disclosures

The following grant information was disclosed by the authors:
Chongqing Municipal Education Commission C: KJQN202104004.
Natural Science Foundation of Chongqing: CSTB2022NSCQ-MSX1632.

## Competing Interests

The authors declare that they have no competing interests.

## Author Contributions

- Zhongya Liu conceived and designed the experiments, analyzed the data, prepared figures and/or tables, and approved the final draft.
- Yunxiao Luo performed the experiments, performed the computation work, authored or reviewed drafts of the article, and approved the final draft.

## Data Availability

The scripts are available in the Supplemental File.

## Supplemental Information

Supplemental information for this article can be found online at http://dx.doi.org/10.7717/peerj-cs.1726#supplemental-information.

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
