# Peer review of "Dynamic coefficient symmetric polynomial-based secure key management scheme for Internet of Things (IoT) networks"

_PeerJ Computer Science, doi:10.7717/peerj-cs.1726_

## Round 0.1 · original submission · Major Revisions

Dear Authors,
Thank you for submitting your manuscript. Please adjust your manuscript to the reviewers' comments. Thank you once again for your manuscript.

Reviewer 1 ·

Basic reporting

Clear and unambiguous, professional English used throughout:
The text appears to be written in clear and unambiguous English. The language used is professional and technical in nature.

Literature references, sufficient field background/context provided:
The text refers to previous works and provides comparisons with other schemes. It demonstrates an understanding of the broader field and discusses relevant prior literature.

Professional article structure, figures, tables. Raw data shared:
It is unclear whether the full article follows a specific structure, as only a specific section is provided. However, the provided section seems to have a logical flow and structure.
The text mentions tables and figures (Table 2, Figure 6, Figure 7, Figure 8, Table 3, Table 4, Table 5, Table 6, Figure 9, Figure 10), but these are not included in the given excerpt, so their quality and appropriateness cannot be assessed.
Raw data is not mentioned or shared in the provided text.

Self-contained with relevant results to hypotheses:
The text appears to be self-contained and focused on presenting the results and comparisons of the proposed key management scheme.
It discusses the resilience to node capture attacks, connectivity rate, and resource overhead of the scheme.

Formal results should include clear definitions of all terms and theorems, and detailed proofs:
The given excerpt does not include specific definitions of terms or theorems, nor does it provide detailed proofs. It mainly presents comparative results and analysis.

Experimental design

Original primary research within Aims and Scope of the journal:
The gtext is within the aims and scope of the Journal..

Research question well defined, relevant & meaningful. It is stated how research fills an identified knowledge gap:
The research question is not explicitly stated in the given excerpt, making it challenging to evaluate its definition.
However, it mentions comparing the proposed key management scheme with other schemes and assessing its resilience to node capture attacks, connectivity rate, and resource overhead. These aspects suggest a research focus on evaluating and improving key management schemes, which could potentially fill a knowledge gap in the field.

Rigorous investigation performed to a high technical & ethical standard:
The text does not provide specific details about the investigation or the methodology employed. Therefore, it is not possible to assess the rigor and technical standard of the investigation.
The text does not mention anything about ethical considerations or ethical standards followed during the research, so it is unclear if the investigation conforms to prevailing ethical standards.

Methods described with sufficient detail & information to replicate:
The given excerpt does not provide detailed information about the methods used in the investigation. Without such details, it is not possible to determine if the methods are described with sufficient detail to be replicated.

Validity of the findings

Provide a clear assessment of the impact and novelty of the research conducted. Explain how the study fills a knowledge gap and contributes to the existing literature. Emphasize the unique aspects or advancements of the research.

Discuss the rationale and benefits of replication studies: If relevant, consider including a section on replication studies and clearly describe the rationale behind replicating the research. Explain how the replication adds value to the literature, such as by validating previous findings or comparing performance metrics.

Ensure that the conclusions are well-stated and directly linked to the original research question. Limit the conclusions to supporting the results obtained from the study. Provide a concise summary of the findings and their implications.

Reviewer 2 ·

Basic reporting

This paper proposes a dynamic coefficient symmetric polynomial key management scheme based on the symmetric polynomial algorithm to solve the IoT security problem, and shows that the proposed scheme significantly improves the resilience against node capture, effectively reduces the communication and storage overheads, and overcomes the security of symmetric polynomial key management scheme. It is interesting and important research issue for IoT networks. However, some descriptions need to be improved and revised.
1. The organization of the paper should be adjusted, and the network environment used by the proposed scheme on Sec. 2.2 should be more clearly stated.
2. Some symbols are not defined clearly, such as SDN and LN on Sec.1.

Experimental design

3. Some important security analyzes such as mutual authentication and authenticated key security (or session key security) are left out.
4. More performance analysis is needed, and it should be clear why the proposed scheme outperforms other related works in terms of efficacy.

Validity of the findings

no comment

Additional comments

no comment

---

## Round 0.2 · accepted · Accept

Dear Dr. Luo,

Thank you for your submission to PeerJ Computer Science.
I am writing to inform you that your manuscript - Dynamic coefficient symmetric polynomial-based secure key management scheme for Internet of Things networks - has been Accepted for publication. Congratulations!

Reviewer 1 ·

Basic reporting

The Authors improved the paper accoring my remarks.

Experimental design

The Authors improved the paper accoring my remarks.

Validity of the findings

The Authors improved the paper accoring my remarks.

Additional comments

The Authors improved the paper accoring my remarks.

Reviewer 2 ·

Basic reporting

no comment

Experimental design

no comment

Validity of the findings

no comment

Additional comments

The authors have revised the descriptions and solved the previous problems. No further comment on this article.